# Association between emergency department attendances, sociodemographic factors and long-term health conditions in the population of Norfolk and Waveney, England: Cross sectional study

**Charlotte E. L. Jones**, **Zillur Rahman Shabuz, Max Bachmann***, **Amanda Burke**, **Julii Brainard**, **Rachel Cullum, Mike Saunders**, **Alice M. Dalton, Oby O. Enwo**, **Nick Steel**

Norwich Medical School, University of East Anglia, Norwich, United Kingdom

* m.bachmann@uea.ac.uk

**Data Availability Statement:** Data availability statement The data are an individual level dataset

## Abstract

### Introduction

Demand for urgent and emergency health care in England has grown over the last decade, for reasons that are not clear. Changes in population demographics may be a cause. This study investigated associations between individuals' characteristics (including socioeconomic deprivation and long term health conditions (LTC)) and the frequency of emergency department (ED) attendances, in the Norfolk and Waveney subregion of the East of England.

### Methods

The study population was people who were registered with 91 of 106 Norfolk and Waveney general practices during one year from 1 April 2022 to 31 March 2023. Linked primary and secondary care and geographical data included each individual's sociodemographic characteristics, and number of ED attendances during the same year and, for some individuals, LTCs and number of general practice (GP) appointments. Associations between these factors and ED attendances were estimated using Poisson regression models.

### Results

1,027,422 individuals were included of whom 57.4% had GP data on the presence or absence of LTC, and 43.1% had both LTC and general practitioner appointment data. In the total population ED attendances were more frequent in individuals aged under five years, (adjusted Incidence Rate Ratio (IRR) 1.25, 95% confidence interval 1.23 to 1.28) compared to 15–35 years; living in more socioeconomically deprived areas (IRR 0.61 (0.60 to 0.63)) for least deprived compared to most deprived, and living closer to the nearest ED. Among individuals with LTC data, each additional LTC was also associated with increased ED

containing information about each individual in the study population, including pseudonymised identify number, social and demographic information, and hospital and GP medical record data. Data are confidential pseudonymised medical records held by the Department of Public Health, Norfolk County Council, and not by the authors who are not authorised to legally distribute or share these data. Data analysis for the study has been done by the authors under a secure digital environment, and subject to the data governance training and requirements, of Norfolk County Council. Data requests may be sent to: Joint Strategic Needs Assessment, Norfolk County Council, County Hall, Martineau Lane, Norwich, Norfolk, NR1 2DH, email JSNA@norfolk.gov.uk.

**Funding:** This research was funded by a grant from Public Health at Norfolk County Council (NCC) https://www.norfolk.gov.uk/. This paper presents independent analysis and research supported by NCC. The views expressed are those of the authors and not necessarily those of NCC, the NHS, or the Department of Health and Social Care. The funders did not have any role in the analysis or interpretation of data or in writing the manuscript.

**Competing interests:** The authors have declared that no competing interests exist.

attendances (IRR 1.16 (1.15 to 1.16)). Among individuals with LTC and GP appointment data, each additional GP appointment was also associated with increased ED attendances (IRR 1.03 (1.026 to 1.027)).

## Conclusions

In the Norfolk and Waveney population, ED attendance rates were higher for young children and individuals living in more deprived areas and closer to EDs. In individuals with LTC and GP appointment data, both factors were also associated with higher ED attendance.

## Introduction

The daily average attendance at major hospital emergency departments (ED) in the 2022/23 financial year in England was 44,400 people [1]. Over the previous decade in England there was a consistently steady annual increase in attendance (13% increase between 2012/13 and 2022/23 [1]), with exceptions during the COVID-19 pandemic (March 2020 to December 2021) [2], and an increase in the average length of time that people spend in ED [3].

Waits of over four hours from ED arrival to departure have become increasingly common. In 2011/12, 5% of patients spent longer than four hours in EDs [2], increasing to 29% in 2022/23 [1]. Despite the number of attendances in 2022/23 being similar to pre-pandemic levels (2% increase in 2022/23 compared to 2019/20 [1]), waiting time performance considerably worsened over this period [3].

Investigations into the determinants of urgent and emergency healthcare utilisation have led to the targeting of interventions at groups with higher risks of ED attendance, with the aim of reducing the frequency of attendances. These interventions have included improving self-management of long-term health conditions [4, 5], improving primary care access for repeat attenders [6], and improving care coordination in older people [7]. Identifying groups with higher ED attendance rates also aids understanding of help-seeking behaviours that lead to ED attendance. For example, progress has been made for children with high attendance rates [8–10].

Predictors of ED attendance previously identified by research are broad, making it difficult to target interventions. Small-scale studies across England have attempted to understand the predictors for ED utilisation using survey data [11, 12], and aggregated general practice (GP) level data, including practice characteristics such as distance from ED, practice list size, and Index of Multiple Deprivation (IMD) based on GP location [13–15]. They found that higher deprivation, lower distance to EDs and higher morbidity (particularly mental health problems) were strongly associated with ED attendance rates [11–15]. Hull and colleagues used linked patient-level secondary and primary care data to investigate whether the association between deprivation and ED attendance in inner urban areas of East London could be explained by clinical variables in GP records in 2014/15. They found that multimorbidity was the strongest clinical predictor for ED attendance [16].

Using individual level data, with over one million linked records from primary and secondary care covering almost the entire population of Norfolk and Waveney, an area with rural, urban, and coastal communities and three acute National Health Service hospital trusts, this study aimed to investigate factors associated with ED attendance. In the whole study population we investigated associations between frequency of ED attendances and deprivation, age, ethnicity, and distance from home to EDs. In a subset of individuals with GP data on presence

or absence of a range of long term conditions (LTCs,) and frequency of GP appointments, we also examined associations between these factors and ED attendances.

## Methods

### Study design and data

The study had a cross-sectional design and was based on the Norfolk and Waveney (N&W) Integrated Care Board's (ICB) developmental linked patient level dataset 2022/23. The dataset was created as an innovative attempt by the N&W ICB to link disparate sources of data relevant to determinants of health, public health and health care needs, for individuals and for local populations, and to inform service commissioning. It was provided temporarily to the authors as part of a broader investigation of urgent and emergency care in this population. The data are not used for payments to service providers or for audits. This dataset contained pseudonymised data on all individuals who were registered at a general practice (GP) in N&W between 1 April 2022 and 31 March 2023. For this analysis, we only included individuals with a residential address within N&W. Of the 106 GPs registered in Norfolk and Waveney [17], 91 were included and provided relevant data on 1,027,422 individuals. Using weighted population estimates for 2022/23 for N&W ICB [17], this dataset covers approximately 90% of the N&W population [18]. It combines data from each individual's GP (primary care), and from hospital trusts, together with age, sex, and ethnicity (coded according to census 2001 ethnic categories [19]), as well as data from the 2019 English indices of multiple deprivation (IMD) decile based on each individual's residential area [20].

The primary care data on each individual included information on presence or absence of a range of LTCs, which were recorded in accordance with the NHS Quality Outcomes Framework for primary care, and the number of GP appointments during the year. General practice data with complete recording of LTCs were only available for 57.4% of the individuals within the dataset. General practice data on numbers of GP appointments during the year were only available for individuals with LTC data, and were available for 75.1% of those individuals. The missing LTC and GP appointment data were not provided to the N&W ICB by participating GPs. We were unable to determine whether these missing data had been collected by GPs or not. Hospital data included the number of ED attendances made by each individual during the year. The dataset contained one record per individual, with no dates of ED attendances or GP appointments, which were recorded as annual counts. The distance (defined by shortest travel time) from each individual's area of residence (census lower layer super output area centroid) to the nearest ED department was calculated using their postal codes.

### Statistical analysis

To investigate individual level predictors of the frequency of ED attendances, we statistically analysed this dataset. We first compared the characteristics of individuals with mean number of ED attendances during the 2022/23 year, using ANOVA F tests. We used multivariable Poisson regression models to estimate associations between the same characteristics and the number of ED visits, adjusted for other covariates [21, 22]. The outcome variable was the number of ED attendances made by an individual between 1 April 2022 and 31 March 2023. The covariates in all models included each individual's age, sex, ethnicity, distance to nearest ED, and residential IMD decile. For individuals with LTC and GP appointment data, these variables were also included in additional models. We used several Poisson regression models, with the following covariates in each model.

- Model 1 included age, sex, ethnicity, IMD decile, distance to nearest ED, and distance squared variables only. IMD decile was coded as a continuous variable, so that the incidence rate ratio (IRR) for IMD is for a difference of one IMD decile. The most deprived decile was coded as 1 and the least deprived decile was coded as 10. Model 1a was identical to Model 1 except that IMD deciles were coded as factor variables, with each decile compared to the most deprived decile as reference category. Model 1 was used to analyse data on the entire study population and, separately, to analyse data restricted to individuals with information on presence or absence of LTCs.

- Models 2–4 were restricted to the subset of individual with LTC data, as follows.

- Model 2 included all the covariates used in Model 1 as well as the number of recorded LTCs.

- Model 3 included all the covariates used in Model 1 as well as nine specific LTCs, each coded as a binary variable.

- Model 4 included all the covariates used in Model 3 and also the number of primary care appointments attended during the year. It was restricted to individuals with LTC and GP appointment data.

To assess the generalisability of the findings of models 2–4 to the total population, we compared the characteristics and ED attendance of participants with and without missing data on long-term health conditions, using chi square and ANOVA F tests.

As sensitivity analyses, we repeated Models 1–4 as mixed models with GP as random effect (that is, with random variation in intercepts among GPs).

R statistical software version 4.1.2 (2021-11-01) [23] was used for all data management and statistical analyses. A 5% significance level was applied.

## Research governance and ethics

Ethical approval was provided by UEA Faculty of Medicine and Health Sciences Research Ethics Subcommittee [ETH2223-17290, 20 March 2023]. The data was held by Norfolk County Council (NCC). Data Protection Impact Assessments were completed in accordance with data governance procedures at NCC for the use of the pseudonymised patient record data. Named UEA researchers delivering the project were provided with NCC laptops to analyse data in the secure NCC data environment. These UEA researchers were accountable to and supervised by the NCC Insight and Analytics Team for the data they accessed and analysed. No data was removed or shared outside of the NCC secure data environment. The study entailed no risk of harm to participants.

## Results

### Results for the total population

The dataset included a total population of 1,027,422 individuals with a residential address within N&W and a recorded sex of male or female. Table 1 shows the characteristics of the study population and the frequency of ED visits associated with each explanatory variable. The population included all age and deprivation bands, with roughly half males and half females. Recorded ethnicity was predominantly White British (59.9%), with 37.9% unknown. All explanatory variables in Table 1 were statistically significantly associated with the mean numbers of ED attendances in univariable models (P<0.001). Children under five years of age had the highest mean number of attendances compared to other age groups. Mean ED attendances were higher in individuals living in the most deprived areas compared to the least deprived.

**Table 1. Characteristics of participants and association between participants' characteristics and emergency department attendance.**

| Variable | No. of individuals (% of column total) | No. who attended at least once (% of row total) | No. of attendances (% of total attendances) | Mean no. attendances | *P |
|---|---|---|---|---|---|
| **Total population** | | | | | |
| Age < = 4 | 37987 (3.7) | 11412 (30.0) | 17400 (7.3) | 0.46 | <0.001 |
| Age 5–14 | 98072 (9.5) | 18071 (18.4) | 24403 (10.3) | 0.25 | |
| Age 15–35 | 214489 (20.9) | 36309 (16.9) | 54377 (22.9) | 0.25 | |
| Age 36–70 | 429852 (41.8) | 56078 (13.0) | 79892 (33.6) | 0.19 | |
| Age >70 | 247022 (24.0) | 39046 (15.8) | 61447 (25.9) | 0.25 | |
| Female | 516682 (50.3) | 82557 (16.0) | 123040 (51.8) | 0.24 | <0.001 |
| Male | 510740 (49.7) | 78359 (15.3) | 114479 (48.2) | 0.22 | |
| White British | 615562 (59.9) | 144081 (23.4) | 214010 (90.1) | 0.35 | <0.001 |
| Ethnicity Not known/missing | 389011 (37.9) | 10847 (2.8) | 15066 (6.3) | 0.04 | |
| Other ethnicity | 22849 (2.2) | 5988 (26.0) | 8443 (3.6) | 0.37 | |
| IMD1 (most deprived) | 93783 (9.1) | 17571 (18.7) | 28668 (12.1) | 0.31 | <0.001 |
| IMD2 | 84812 (8.3) | 14825 (17.5) | 22635 (9.5) | 0.27 | |
| IMD3 | 91128 (8.9) | 15843 (17.4) | 24373 (10.3) | 0.27 | |
| IMD4 | 152868 (14.9) | 23684 (15.5) | 34881 (14.7) | 0.23 | |
| IMD5 | 165166 (16.1) | 25395 (15.4) | 36957 (15.6) | 0.22 | |
| IMD6 | 144707 (14.1) | 21888 (15.1) | 31618 (13.3) | 0.22 | |
| IMD7 | 93324 (9.1) | 13438 (14.4) | 19023 (8.0) | 0.20 | |
| IMD8 | 86008 (8.4) | 12239 (14.2) | 17235 (7.3) | 0.20 | |
| IMD9 | 67145 (6.5) | 9525 (14.2) | 13205 (5.6) | 0.20 | |
| IMD10 (least deprived) | 48475 (4.7) | 6508 (13.4) | 8924 (3.8) | 0.18 | |
| Total records | 1027422 (100) | 160916 (15.7) | 237519 (100) | 0.23 | |
| **Individuals with long term condition data** | | | | | |
| Respiratory condition** | 154150 (26.1) | 30936 (20.1) | 50643 (32.7) | 0.33 | <0.001 |
| No respiratory condition | 435414 (73.9) | 68831 (15.8) | 104298 (67.3) | 0.24 | |
| Pre-diabetes | 75117 (12.7) | 14422 (19.2) | 22312 (14.4) | 0.30 | <0.001 |
| No pre-diabetes | 514447 (87.3) | 85345 (16.6) | 132629 (85.6) | 0.26 | |
| Diabetes | 77624 (13.2) | 15073 (19.4) | 25223 (16.3) | 0.32 | <0.001 |
| No diabetes | 511940 (86.8) | 84694 (16.5) | 129718 (83.7) | 0.25 | |
| Heart disease | 72417 (12.3) | 15486 (21.4) | 27247 (17.6) | 0.38 | <0.001 |
| No heart disease | 517147 (87.7) | 84281 (16.3) | 127694 (82.4) | 0.25 | |
| Atrial fibrillation | 47510 (8.1) | 10662 (22.4) | 18981 (12.3) | 0.40 | <0.001 |
| No atrial fibrillation | 542054 (91.9) | 89105 (16.4) | 135960 (87.7) | 0.25 | |
| Kidney disease | 62179 (10.5) | 11634 (18.7) | 19668 (12.7) | 0.32 | <0.001 |
| No kidney disease | 527385 (89.5) | 88133 (16.7) | 135273 (87.3) | 0.26 | |
| Depression | 179274 (30.4) | 35824 (20.0) | 58979 (38.1) | 0.33 | <0.001 |
| No depression | 410290 (69.6) | 63943 (15.6) | 95962 (61.9) | 0.23 | |
| Hypertension | 193559 (32.8) | 33111 (17.1) | 52098 (33.6) | 0.27 | <0.001 |

*(Continued)*

**Table 1.** (Continued)

| Variable | No. of individuals (% of column total) | No. who attended at least once (% of row total) | No. of attendances (% of total attendances) | Mean no. attendances | *P |
|---|---|---|---|---|---|
| No hypertension | 396005 (67.2) | 66656 (16.8) | 102843 (66.4) | 0.26 | |
| Stroke | 18422 (3.1) | 4133 (22.4) | 7409 (4.8) | 0.40 | <0.001 |
| No stroke | 571142 (96.9) | 95634 (16.7) | 147532 (95.2) | 0.26 | |
| Total individuals with long term condition data | 589564 (100) | 99767 (16.9) | 154941 (100) | 0.29 | |

*P values from ANOVA F-test comparing mean no. attendances between categories

** Chronic respiratory conditions including asthma chronic obstructive pulmonary disease and chronic respiratory disease

Results for the multivariable analysis that included the total population are reported in Table 2. All covariates were statistically significantly associated with the rate of yearly hospital attendances. Individuals under five years old had a 25% higher rates of hospital attendances compared to those aged 15 to 35 years (adjusted incidence rate ratio (IRR) 1.25, 95% confidence interval (CI) 1.23 to 1.28). Both the age groups of five to 14 years and over 70 years had approximately 8% lower ED attendance rates (IRR 0.92), while those aged 36 to 70 years had 28% lower attendance rates (IRR 0.72, 95% CI 0.71 to 0.73). Males had 2% higher rates of ED attendances compared to females (IRR 1.02, 95% CI 1.01 to 1.02). Individuals whose ethnicity was not known or missing had a much lower rate of ED attendances (IRR 0.11, 95% CI 0.11 to 0.12) than the white British category. For each unit increase in IMD decile, there was 5% (IRR 0.95, 95% CI 0.95 to 0.95) decrease in the rate of hospital attendances. For every 10km increase in distance to the nearest hospital, the rate of ED attendances decreased (IRR 0.86, 95%CI 0.85 to 0.87). However, the deterrent effect of distance on ED attendances decreased with increasing distance (IRR>1 for distance squared). With IMD decile modelled as a categorical variable, ED attendance rates decreased steadily with increasing IMD decile (that is, with decreasing deprivation). Individuals in the least deprived decile (IMD10) had rates of ED attendances 39% lower compared to those in the most deprived decile (IMD1) (IRR = 0.61, 95%CI 0.60 to 0.63).

## Results for individuals with long term condition and general practice appointment data

Of the total population, 589,564 (57.4%) individuals had LTC data and were included in analyses which used LTC data. Characteristics of individuals with and without LTC data are compared in Table 3. Those with missing LTC data had higher probabilities of being younger and male, more deprived and not identified as white British and had, on average, fewer ED visits and lived further from EDs.

Results of multivariable models 1–4, including only individuals who had LTC data, are shown in Table 4. In Model 1, (without LTC covariates) the IRRs associated with each covariate were similar to the equivalent Model 1 results for the whole study population, except that the IRR for age under 5 was greater (IRR 1.97 compared to 1.25 in Table 2). In Model 2 (including the number of LTCs as covariate) there was a 16% (IRR 1.16, 95% CI 1.15 to 1.16) increase in the rate of ED attendances for each additional condition. This adjustment for number of LTCs decreased the IRRs associated with age 36–70 (IRR 0.53 in Model 2 compared to

**Table 2. Association between sociodemographic factors and number of emergency department attendances in the total population (n = 1,027,422): Poisson regression model.**

| Number of participants | 1,027,422 | | |
|---|---|---|---|
| Variables | IRR | 95% CI | p-value |
| Age < = 4 | 1.25 | (1.23, 1.28) | <0.001 |
| Age 5–14 | 0.92 | (0.90, 0.93) | <0.001 |
| Age 15–35 (reference) | – | – | – |
| Age 36–70 | 0.72 | (0.71, 0.73) | <0.001 |
| Age >70 | 0.92 | (0.91, 0.94) | <0.001 |
| Female (ref) | – | – | – |
| Male | 1.02 | (1.01, 1.02) | <0.001 |
| White British (ref) | – | – | – |
| Not known/missing | 0.11 | (0.11, 0.12) | <0.001 |
| Other ethnicity | 0.98 | (0.96, 1.00) | 0.094 |
| Distance (10km) | 0.86 | (0.85, 0.87) | <0.001 |
| Distance square (100km$^2$) | 1.03 | (1.02, 1.03) | <0.001 |
| IMD (continuous per decile) | 0.95 | (0.95, 0.95) | <0.001 |
| IMD deciles* | | | |
| IMD 1 (reference, most deprived) | – | – | – |
| IMD 2 | 0.89 | (0.87, 0.91) | <0.001 |
| IMD 3 | 0.89 | (0.87, 0.90) | <0.001 |
| IMD 4 | 0.80 | (0.79, 0.81) | <0.001 |
| IMD 5 | 0.77 | (0.76, 0.79) | <0.001 |
| IMD 6 | 0.75 | (0.74, 0.77) | <0.001 |
| IMD 7 | 0.70 | (0.69, 0.71) | <0.001 |
| IMD 8 | 0.68 | (0.67, 0.70) | <0.001 |
| IMD 9 | 0.65 | (0.64, 0.66) | <0.001 |
| IMD 10 (least deprived) | 0.61 | (0.60, 0.63) | <0.001 |

IRR Incidence rate ratio, CI confidence interval, IMD Index of Multiple Deprivation

*Model 1a: IRRs for IMD deciles adjusted for all other covariates

of Multiple Deprivation

*Model 1a: IRRs for IMD deciles adjusted for all other covariates

0.63 in Model 1) and >70 (IRR 0.47 compared to 0.76). In Model 3 (including specific LTCs) every condition was associated with increased rates of ED attendances, with atrial fibrillation, stroke, depression, and heart disease having the highest IRRs. In Model 4, ED attendances increased by 3% (IRR 1.03, 95% CI 1.03 to 1.03) for every extra primary care appointment per year. The IRRs for the other variables in Model 4 are similar to those in Model 3, except that in Model 4 the small sex difference changed direction, the IRRs for diabetes and pre-diabetes were smaller, and for hypertension the IRR was 1.00 (95% CI 0.99 to 1.02).

Models 1–4 implemented as mixed models with GP as random effect produced very similar results to the primary analysis (S1 and S2 Tables in S1 File).

## Discussion

This study shows that the frequency with which individuals in N&W attended EDs was dependent on factors previously known to be associated with poor health, including socioeconomic deprivation. Sex and ethnic differences were relatively small, although missing ethnicity data may have impacted those results. The growing burden on emergency services has been partly

**Table 3. Characteristics of individuals with and without long term condition data.**

| Variables | Total participants | | With long term condition data | | Without long term condition data | | |
|---|---|---|---|---|---|---|---|
| **Categorical** | n | % | n | % | n | % | P* |
| Age < = 4 | 37987 | 3.7 | 1543 | 0.3 | 36444 | 8.3 | <0.001 |
| Age 5–14 | 98072 | 9.5 | 14350 | 2.4 | 83722 | 19.1 | |
| Age 15–35 (ref) | 214489 | 20.9 | 78843 | 13.4 | 135646 | 31.0 | |
| Age 36–70 | 429852 | 41.8 | 274356 | 46.5 | 155496 | 35.5 | |
| Age >70 | 247022 | 24.0 | 220472 | 37.4 | 26550 | 6.1 | |
| Male | 510740 | 49.7 | 279970 | 47.5 | 230770 | 52.7 | <0.001 |
| Female | 516682 | 50.3 | 309594 | 52.5 | 207088 | 47.3 | |
| White British | 615562 | 59.9 | 392176 | 66.5 | 223386 | 51.0 | <0.001 |
| Ethnicity not known/missing | 389011 | 37.9 | 189059 | 32.1 | 199952 | 45.7 | |
| Other ethnicity | 22849 | 2.2 | 8329 | 1.4 | 14520 | 3.3 | |
| Total | 1027422 | 100 | 589564 | 100 | 437858 | 100 | |
| **Continuous** | mean | SD | mean | SD | mean | SD | P** |
| Index of multiple deprivation | 5.13 | 2.48 | 5.24 | 2.43 | 4.99 | 2.55 | <0.001 |
| Distance from ED | 18.75 | 12.36 | 19.10 | 12.27 | 18.29 | 12.45 | <0.001 |
| No. ED visits | 0.23 | 0.74 | 0.26 | 0.85 | 0.19 | 0.56 | <0.001 |

* p value from chi square test

** P value from ANOVA F-test. ED emergency department

attributed to the increasing proportions of the UK population that are older. In this study, individuals aged over 70 years had lower rates of ED attendances than any age categories younger than 36 years, but they made up 25.9% of all attendances. Children aged under five years had the highest rates of ED attendances, and were responsible for 7.3% of total attendances. However young children do not necessarily place a disproportionate burden on EDs. Previous studies have shown that children spend less time in EDs compared to the elderly [24], and are more likely to be classified as non-urgent compared to other age groups [25]. ED attendance rates increased steadily with deprivation. Greater distance from home to the nearest ED was associated with lower attendance rates, which presumably reflects less accessibility of care. However, this effect was attenuated at greater distances.

Analysis of data restricted to individuals with LTC data showed that LTCs were associated with more frequent ED attendances. This is to be expected as LTCs are known to be associated with increased risk of medical emergencies. The results suggest that the co-existence of additional LTCs increased the risk of ED attendance multiplicatively, whether LTCs were modelled as single conditions or as number of LTCs. Adjustment for long-term health conditions reduced the incidence ratios for those aged 36–70 and those over 70 (Models 2–4 compared to Model 1 in Table 4), suggesting that in those two age groups ED attendance rates are largely explained by their higher prevalence of long-term health conditions. Frequency of GP appointments was also associated with greater ED attendance, presumably because it is also an indicator of ill health.

While previous research has attributed the association between deprivation and ED attendance to higher levels of multimorbidity in more deprived populations [13–16], in our study adjustment for LTCs did not change the association between deprivation and ED attendance rates (models 2–4 compared to Model 1 in Table 4). This suggests that LTCs may not be the main reason that more deprived people attend EDs more often. However this inference should be cautious, given the limitations of our LTC data.

**Table 4. Association between explanatory variables and frequency of emergency department visit in individuals with data on long term conditions or general practice attendances: Poisson regression models.**

| | Model 1 | | | Model 2 | | | Model 3 | | | Model 4 | | |
|---|---|---|---|---|---|---|---|---|---|---|---|---|
| | (Without long term conditions as covariate) | | | (No. long term conditions as continuous variable) | | | (Each long term condition as binary variable) | | | (No. primary care appointment and each long term condition as covariates) | | |
| No. participants | | 589,564 | | | 589,564 | | | 589,564 | | | 442,900 | |
| Variables | IRR | 95% CI | p-value | IRR | 95% CI | p-value | IRR | 95% CI | p-value | IRR | 95% CI | p-value |
| Age < = 4 | 1.97 | (1.88, 2.08) | <0.001 | 2.13 | (2.02, 2.25) | <0.001 | 2.08 | (1.98, 2.12) | <0.001 | 1.93 | (1.82, 2.03) | <0.001 |
| Age 5–14 | 0.96 | (0.93, 0.99) | 0.003 | 1.01 | (0.99, 1.04) | 0.324 | 1.06 | (1.03, 1.09) | <0.001 | 1.15 | (1.12, 1.19) | <0.001 |
| Age 15–35 (ref) | | | – | – | – | – | – | – | – | – | – | – |
| Age 36–70 | 0.63 | (0.62, 0.64) | <0.001 | 0.53 | (0.53, 0.54) | <0.001 | 0.60 | (0.59, 0.61) | <0.001 | 0.60 | (0.58, 0.60) | <0.001 |
| Age >70 | 0.76 | (0.75, 0.77) | <0.001 | 0.47 | (0.47, 0.48) | <0.001 | 0.62 | (0.61, 0.63) | <0.001 | 0.68 | (0.67, 0.70) | <0.001 |
| Female (ref) | | | – | – | – | – | – | – | – | – | – | – |
| Male | 0.97 | (0.96, 0.98) | <0.001 | 0.95 | (0.94, 0.96) | <0.001 | 0.96 | (0.95, 0.97) | <0.001 | 1.03 | (1.02, 1.04) | <0.001 |
| White British (ref) | | | – | – | – | – | – | – | – | – | – | – |
| Not known/missing | 0.15 | (0.14, 0.15) | <0.001 | 0.16 | (0.15, 0.16) | <0.001 | 0.15 | (0.15, 0.15) | <0.001 | 0.25 | (0.25, 0.26) | <0.001 |
| Other ethnicity | 1.00 | (0.96, 1.04) | 0.97 | 1.03 | (0.99, 1.06) | 0.142 | 1.04 | (1.01, 1.08) | 0.021 | 1.09 | (1.05, 1.13) | <0.001 |
| IMD (continuous per decile) | 0.95 | (0.94, 0.95) | <0.001 | 0.95 | (0.95, 0.96) | <0.001 | 0.95 | (0.95, 0.96) | <0.001 | 0.96 | (0.95, 0.96) | <0.001 |
| Distance (10km) | 0.85 | (0.84, 0.86) | <0.001 | 0.85 | (0.84, 0.87) | <0.001 | 0.85 | (0.84, 0.86) | <0.001 | 0.82 | (0.81, 0.83) | <0.001 |
| Distance square (100km$^2$) | 1.03 | (1.02, 1.03) | <0.001 | 1.03 | (1.02, 1.03) | <0.001 | 1.03 | (1.03, 1.03) | <0.001 | 1.03 | (1.03, 1.04) | <0.001 |
| Number of long-term health conditions | | | | 1.16 | (1.15, 1.16) | <0.001 | | | | | | |
| Respiratory | | | | | | | 1.24 | (1.23, 1.26) | <0.001 | 1.18 | (1.17, 1.19) | <0.001 |
| Pre-diabetes | | | | | | | 1.08 | (1.07, 1.10) | <0.001 | 0.92 | (0.90, 0.93) | <0.001 |
| Diabetes | | | | | | | 1.19 | (1.17, 1.21) | <0.001 | 1.06 | (1.05, 1.08) | <0.001 |
| Heart disease | | | | | | | 1.37 | (1.35, 1.39) | <0.001 | 1.44 | (1.42, 1.46) | <0.001 |
| Atrial fibrillation | | | | | | | 1.48 | (1.45, 1.50) | <0.001 | 1.41 | (1.39, 1.44) | <0.001 |
| Kidney disease | | | | | | | 1.12 | (1.10, 1.14) | <0.001 | 1.15 | (1.13, 1.17) | <0.001 |
| Depression | | | | | | | 1.38 | (1.36, 1.39) | <0.001 | 1.27 | (1.26, 1.29) | <0.001 |
| Hypertension | | | | | | | 1.06 | (1.05, 1.07) | <0.001 | 1.00 | (0.99, 1.02) | 0.457 |
| Stroke | | | | | | | 1.38 | (1.34, 1.41) | <0.001 | 1.41 | (1.38, 1.45) | <0.001 |
| Number of primary care appointments | | | | | | | | | | 1.03 | (1.03, 1.03) | <0.001 |

IRR Incidence rate ratio, CI confidence interval

Our findings broadly agree with previous research in the UK about correlates with higher ED attendance rates. Hull and colleagues found that higher attendances were linked with higher GP consultation rates [14]. Other studies have found that ED attendances rose with increasing co-occurrence of long-term health conditions [12, 13, 16], and that attendance rates were higher in individuals living in the most deprived English areas [13–16]. In an analysis of data available in 2008 for the West Midlands in central England (population 5.4 million), Rudge and colleagues [15] found declining ED attendance rates with increasing travel distance, although this relationship was modified by deprivation. Hull and colleagues [14] and Giebel and colleagues [11] also reported declining attendance rates with increased distance to home residence. Previous research is less consistent about which age groups have the highest utilisation rates of EDs. In a retrospective cohort analysis of London residents, Hull and colleagues found that crude and adjusted ED attendance rates were higher for persons aged 70 years and over than for persons under age five years [14]. Conversely, in a cross-sectional study of all English GPs, Scantlebury and colleagues [13] found that the proportion of the population aged

under five years in the same geographic area was positively related to ED attendances. Differences in the findings of these previous studies may partly be attributed to differences in the range of confounding variables they were able to account for.

A key strength of this study is the large dataset, linking primary and secondary care data together with sociodemographic and geographical data, enabling us to investigate the independent effects of many interlinked variables in a geographically defined population. The data included almost all of the population residing within the study area, and the large sample size provided high statistical power and precision. This is the first external use of the N&W ICBs developmental linked patient level dataset. Such analyses have the potential for improving patient care and has created opportunities for collaboration between local authorities, the National Health Service and academic researchers.

However, the dataset is still developmental and subject to data quality issues. Of the 106 GPs in N&W, 15 opted out of sharing any of their data. The dataset also had many missing values, particularly for ethnicity (37.9% missing or not known), diagnosis codes for LTCs (42.6% missing) and GP appointments (56.9% missing). This level of missing information is common across NHS healthcare datasets [26, 27], and has particularly been highlighted in the use of linked datasets where GPs may use different coding procedures [28]. To our knowledge this is the first use of individual level data to analyse the associations between ED attendance and specific long-term health conditions in a large geographically defined English population. However, the large proportion of participants with missing data on long-term health conditions and GP appointments, with more missing data at younger ages, raising doubts about the accuracy of these variables and the analyses that used them. The results of the analyses involving long-term health conditions are therefore not necessarily generalisable to the whole study population. For future epidemiological research linking sociodemographic, primary and secondary care data for geographically defined populations in Norfolk and Waveney, and the United Kingdom more generally, this study shows that it is a priority to enable more complete linkage of informative primary care data about individuals' health states.

Another limitation of the study is its cross-sectional design. The dataset aggregated events over a single twelve-month period, which precludes identification of earlier causes and later effects. For instance, it is plausible that greater access to and use of primary care could have led to less ED use, but it is also plausible that some primary care visits might have followed from ED visits. Our finding that GP appointment rates were associated with higher ED attendance rates may also be due to higher rates of GP appointments for those in poor health.

## Conclusion

This study demonstrates what can be learned from individual patient record data, its use for informing the evidence base about the challenges faced in recent years by urgent and emergency care services in England, and the caveats that apply. Average ED attendance rates during 2022/23 were greater in people aged under five years; living in more socioeconomically deprived areas; living closer to ED; with a long-term medical condition; or with more GP attendances during the same year. Our findings align with the current evidence and further add to the evidence base, particularly about how different long-term health conditions affect ED attendance frequency. The findings support the use of interventions and policy aimed at preventing and managing LTCs, for example the National Health Service Prevention Programme in England [29]. Future research could investigate characteristics of frequent attendees using similar methods and linked data.

## Supporting information

**S1 File. Poisson regression mixed models with general practice as random effect.**
(DOCX)

## Author Contributions

**Conceptualization:** Max Bachmann, Amanda Burke, Nick Steel.

**Data curation:** Zillur Rahman Shabuz, Max Bachmann.

**Formal analysis:** Zillur Rahman Shabuz, Max Bachmann.

**Funding acquisition:** Max Bachmann, Nick Steel.

**Investigation:** Zillur Rahman Shabuz, Max Bachmann.

**Methodology:** Charlotte E. L. Jones, Zillur Rahman Shabuz, Max Bachmann, Nick Steel.

**Project administration:** Amanda Burke, Rachel Cullum, Nick Steel.

**Supervision:** Amanda Burke, Nick Steel.

**Validation:** Zillur Rahman Shabuz.

**Writing – original draft:** Charlotte E. L. Jones, Max Bachmann, Julii Brainard.

**Writing – review & editing:** Charlotte E. L. Jones, Zillur Rahman Shabuz, Max Bachmann, Amanda Burke, Julii Brainard, Rachel Cullum, Mike Saunders, Alice M. Dalton, Oby O. Enwo, Nick Steel.

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
