## [Decision Letter · Decision Letter 0]

30 Jan 2024

PONE-D-23-42940Associations between emergency department attendances and long-term health conditions and sociodemographic factors in the population of Norfolk and Waveney, England: Cross sectional studyPLOS ONE

Dear Dr. Bachmann,

Thank you for submitting your manuscript to PLOS ONE. After careful consideration, we feel that it has merit but does not fully meet PLOS ONE’s publication criteria as it currently stands. Therefore, we invite you to submit a revised version of the manuscript that addresses the points raised during the review process.

We look forward to receiving your revised manuscript.

Kind regards,

Sreeram V. Ramagopalan

Academic Editor

PLOS ONE

2. For studies involving third-party data, we encourage authors to share any data specific to their analyses that they can legally distribute. PLOS recognizes, however, that authors may be using third-party data they do not have the rights to share. When third-party data cannot be publicly shared, authors must provide all information necessary for interested researchers to apply to gain access to the data. (https://journals.plos.org/plosone/s/data-availability#loc-acceptable-data-access-restrictions)

a) A description of the data set and the third-party source

b) If applicable, verification of permission to use the data set

c) Confirmation of whether the authors received any special privileges in accessing the data that other researchers would not have

d) All necessary contact information others would need to apply to gain access to the data

3. We notice that your supplementary tables are included in the manuscript file. Please remove them and upload them with the file type 'Supporting Information'. Please ensure that each Supporting Information file has a legend listed in the manuscript after the references list.

Reviewers' comments:

Reviewer's Responses to Questions

**Comments to the Author**

1. Is the manuscript technically sound, and do the data support the conclusions?

Reviewer #1: Partly

2. Has the statistical analysis been performed appropriately and rigorously? 

Reviewer #1: Yes

3. Have the authors made all data underlying the findings in their manuscript fully available?

Reviewer #1: Yes

4. Is the manuscript presented in an intelligible fashion and written in standard English?

Reviewer #1: Yes

5. Review Comments to the Author

Reviewer #1: Dear Editor,

Many thanks for the opportunity to read the manuscript titled “Associations between emergency department attendances and long-term health conditions and sociodemographic factors in the population of Norfolk and Waveney, England: Cross sectional study.”

The manuscript reports an analysis of a patient level administrative dataset containing linked primary and secondary care records. The aim is to determine factors associated with attendance to the emergency department (ED) in a geographically diverse area. The authors report two analysis, one containing the whole study population (which represents almost the entire population of Norfolk and Waveney) and a subset with diagnostic data.

The authors conclude that long term health conditions, age, socioeconomic deprivation and distance to services are associated with ED attendance.

Although I don’t disagree with the methodology used, I think much clearer reporting is needed. My full comments are below.

Major comments:

Methods:

1.- Please clarify in the methods that there were no event dates in the dataset.

2.- What is the purpose of the N&W ICB dataset – is it also linked to payments or is it an audit dataset?

3.- Primary care data were recorded in accordance to QOF. QOF is a very good dataset to determine prevalence of disease in the population since it is incentivised by payment. Therefore I find it difficult to understand the high levels of missingness. Is it more that individuals without long term health conditions appear to have missing data, is it a data linkage issue or data was not collected at source?

4. Given that temporality cannot be assessed wouldn’t it be more accurate to rephrase Long-term health condition with Chronic health conditions.

5.- Clarify whether respiratory disease includes Asthma / COPD only or whether it includes acute respiratory infections.

Results:

6. Model 2-4 are reporting results for a completely different population to those included in model 1. My interpretation is that model 2-4 report results for a population with hihg burden of disease. Particularly the children included in this subpopulation must be significantly more poorly than the general population to have acquired a long-term health condition. This is clear to see in table 3. Therefore, results should be results should be reported separately.

Whole study population analysis should be reported completely separately from the analysis including only population with long term health conditions. Fixed model reported in Table S2 should be moved into the main manuscript (table 2) so that the comparisons the authors highlight in the discussion are made in the results.

7.- The first section of the results should report all results for the whole population (demographics as well as model 1). Why wasn’t the number of primary care appointments modelled as part of model 1? Could be a good compromise for ascertaining the impact of multimorbidity via higher attendance to GP practice in the whole population.

Abstract

8.- Abstract reports results from model 1, model 2 and model 4 without clarification that there is more than one analysis. Results should be reported for model 1 and than it should be made clear that in a subset with diagnostic data X was the impact of long term conditions. Similarly including primary care appointments into model 1 would strengthen this analysis (although it still wouldn’t address cause and effect).

9-Abstract conclusions lead with - longer term health conditions. Given that these results might not be representative of the population in Norfolk and Waveney how do the authors justify this conclusion.

Discussion

10.- Missing data for ethnicity is common in NHS datasets however the high level of diagnosis data missing in an administrative dataset seems high, particularly if it is linked to payments. GPs tend to have very good recording of QOF metrics, which makes it a very good dataset for establishing prevalence of disease in the population.

Tables

11. Table 1 is confusing with proportions being calculated in columns and rows and the addition at the bottom of a new denominator (i.e. those with diagnosis data). Here the results should also be split so that the full population analysis is reported first and then the subpopulation is reported separately.

6. PLOS authors have the option to publish the peer review history of their article (what does this mean?). If published, this will include your full peer review and any attached files.

Reviewer #1: No

---

## [Author Response · Author response to Decision Letter 0]

26 Mar 2024

Thank you for your invitation to resubmit this manuscript after revising it in the light of the reviewers’ comments. We thank the reviewers for their helpful comments, which we believe have greatly improved the article. Our responses to each comment are detailed, as bullet points, as follows. 

Comments to the Author

Reviewer #1: 

Major comments:

Methods:

1.- Please clarify in the methods that there were no event dates in the dataset.

• Response: We have added to the second paragraph of Methods. “The dataset contained one record per individual, with no dates of ED attendances or general practice appointments.”

2.- What is the purpose of the N&W ICB dataset – is it also linked to payments or is it an audit dataset?

• Response: We have added to the first paragraph of Methods, “The dataset was created as an innovative attempt by the N&W ICB to link disparate sources of data relevant to determinants of health, public health and health care needs, for individuals and for local populations, and to inform service commissioning. It was provided temporarily to the authors as part of a broader investigation of influences on urgent and emergency care in this population. The data are not used for payments to service providers or for audits.”

3.- Primary care data were recorded in accordance to QOF. QOF is a very good dataset to determine prevalence of disease in the population since it is incentivised by payment. Therefore I find it difficult to understand the high levels of missingness. Is it more that individuals without long term health conditions appear to have missing data, is it a data linkage issue or data was not collected at source?

• Response: As we have added to the second paragraph of Methods, “The missing long term condition and general practice appointment data were not provided to the N&W ICB by some participating general practices. We were unable to determine whether these missing data had been collected by general practices or not.”

4. Given that temporality cannot be assessed wouldn’t it be more accurate to rephrase Long-term health condition with Chronic health conditions.

• Response: Having a ‘long term condition’ does not necessarily mean that an individual has already had it for a long time. “Long term conditions” and “chronic conditions” usually have the same meaning, referring a range of conditions that tend to last for decades and cannot be cured. We would prefer to continue to use the term ‘long term conditions’ because that is nowadays the preferred terminology used by the National Health Service (see for example , https://www.england.nhs.uk/ourwork/clinical-policy/ltc/) and by many other researchers in this field. 

5.- Clarify whether respiratory disease includes Asthma / COPD only or whether it includes acute respiratory infections.

• We have added a footnote to Table 1 defining ‘Respiratory conditions” as “Chronic respiratory conditions including asthma and chronic obstructive pulmonary disease”

Results:

6. Model 2-4 are reporting results for a completely different population to those included in model 1. My interpretation is that model 2-4 report results for a population with hihg burden of disease. Particularly the children included in this subpopulation must be significantly more poorly than the general population to have acquired a long-term health condition. This is clear to see in table 3. Therefore, results should be results should be reported separately. Whole study population analysis should be reported completely separately from the analysis including only population with long term health conditions. 

• Response: We have thoroughly revised the entire manuscript (Abstract, Methods, Results, Tables, and Discussion) to separate the analysis and reporting of the whole population from the analysis and reporting of the subgroup with long term conditions and general practice appointment data. 

Fixed model reported in Table S2 should be moved into the main manuscript (table 2) so that the comparisons the authors highlight in the discussion are made in the results.

• Response: Also in response to the previous comment we have divided the original Table 2 into a new Table 2 (reporting model 1 results for the total population) and a new Table 4 (reporting models 1-4 for the population with LTA data, and including the model 1 fixed effect results from the original Table S2 as recommended). 

7.- The first section of the results should report all results for the whole population (demographics as well as model 1). Why wasn’t the number of primary care appointments modelled as part of model 1? Could be a good compromise for ascertaining the impact of multimorbidity via higher attendance to GP practice in the whole population.

• Response: We have revised the first section of Results to report only results for the whole population. General practice appointment data were only reported for individuals with LTC data. We have added to Methods, Study design and data: “General practice data on numbers of general practice appointments during the year were only available for individuals with LTC data, and were available for 75.1% of those individuals.”

Abstract

8.- Abstract reports results from model 1, model 2 and model 4 without clarification that there is more than one analysis. Results should be reported for model 1 and than it should be made clear that in a subset with diagnostic data X was the impact of long term conditions. Similarly including primary care appointments into model 1 would strengthen this analysis (although it still wouldn’t address cause and effect).

• Response: We have changed the Abstract accordingly. As stated above, we were unable to include general practices appointments in Model 1 for the total population because of missing data. 

9-Abstract conclusions lead with - longer term health conditions. Given that these results might not be representative of the population in Norfolk and Waveney how do the authors justify this conclusion.

• Response: We have changed the Abstract conclusions to: “In the Norfolk and Waveney population, ED attendance rates were higher for young children and individuals living in more deprived areas and closer to EDs. In individuals with LTC and general practice appointment data, both factors were also associated with higher ED attendance.”

Discussion

10.- Missing data for ethnicity is common in NHS datasets however the high level of diagnosis data missing in an administrative dataset seems high, particularly if it is linked to payments. GPs tend to have very good recording of QOF metrics, which makes it a very good dataset for establishing prevalence of disease in the population.

• Response: In Discussion, Strengths and Limitations, we have added, “For future epidemiological research linking sociodemographic, primary and secondary care data for geographically defined populations in Norfolk and Waveney, and the United Kingdom more generally, this study shows that it is a priority to enable more complete linkage of informative primary care data about individuals’ health states.”

Tables

11. Table 1 is confusing with proportions being calculated in columns and rows and the addition at the bottom of a new denominator (i.e. those with diagnosis data). Here the results should also be split so that the full population analysis is reported first and then the subpopulation is reported separately.

• Response: We have subdivided Table 1 to separate results for the total population and for those with LTC data. We have edited the column headings to clarify whether the denominators are row or column totals.

---

## [Editor Report · Decision Letter 1]

23 Apr 2024

Association between emergency department attendances, sociodemographic factors and long-term health conditions in the population of Norfolk and Waveney, England: Cross sectional study

PONE-D-23-42940R1

Dear Dr. Bachmann,

We’re pleased to inform you that your manuscript has been judged scientifically suitable for publication and will be formally accepted for publication once it meets all outstanding technical requirements.

Kind regards,

Sreeram V. Ramagopalan

Academic Editor

PLOS ONE
---

## [Editor Report · Acceptance letter]

26 Apr 2024

PONE-D-23-42940R1 

PLOS ONE

Dear Dr. Bachmann, 

I'm pleased to inform you that your manuscript has been deemed suitable for publication in PLOS ONE. Congratulations! Your manuscript is now being handed over to our production team.

Kind regards, 

on behalf of

Dr. Sreeram V. Ramagopalan 

Academic Editor

PLOS ONE